# Health Benefits of β-Hydroxy-β-Methylbutyrate (HMB) Supplementation in Addition to Physical Exercise in Older Adults: A Systematic Review with Meta-Analysis

**DOI:** 10.3390/nu11092082

**Published:** 2019-09-03

**Authors:** Javier Courel-Ibáñez, Tomas Vetrovsky, Klara Dadova, Jesús G. Pallarés, Michal Steffl

**Affiliations:** 1Human Performance and Sports Science Laboratory, Faculty of Sport Sciences, University of Murcia, 30100 Murcia, Spain; 2Faculty of Physical Education and Sport, Charles University, 16252 Prague, Czech Republic

**Keywords:** nutrition, resistance training, leucine, elderly, sarcopenia, neuromuscular function

## Abstract

Both regular exercise training and beta-hydroxy-beta-methylbutyrate (HMB) supplementation are shown as effective treatments to delay or reverse frailty and reduce cognitive impairment in older people. However, there is very little evidence on the true benefits of combining both strategies. The aim of this meta-analysis was to quantify the effects of exercise in addition to HMB supplementation, on physical and cognitive health in older adults. Data from 10 randomized controlled trials (RCTs) investigating the effect of HMB supplementation and physical function in adults aged 50 years or older were analyzed, involving 384 participants. Results showed that HMB supplementation in addition to physical exercise has no or fairly low impact in improving body composition, muscle strength, or physical performance in adults aged 50 to 80 years, compared to exercise alone. There is a gap of knowledge on the beneficial effects of HMB combined with exercise to preserve cognitive functions in aging and age-related neurodegenerative diseases. Future RCTs are needed to refine treatment choices combining HMB and exercises for older people in particular populations, ages, and health status. Specifically, interventions in older adults aged 80 years or older, with cognitive impairment, frailty, or limited mobility are required.

## 1. Introduction

Evidence supports the fact that the combination of muscle strength training and protein supplementation stands as the most effective and easiest intervention to delay or reverse frailty in primary care [1,2] and emerges as a plausible treatment to reduce functional and cognitive impairment in older adults [3].

There is increasing evidence that tailored multicomponent exercise programs benefit both physical and cognitive health in frail older people [4,5,6,7,8], to the extent that it is being considered mandatory for community-dwelling and institutionalized people [9]. However, the level of regular physical activity and resistance training of older ages is likely to be much lower than recommended [10]. Physical inactivity in older adults represents a serious health risk as it contributes to the onset of muscle mass and function decline, which—sometimes irremediably—leads to frailty and derived short-term and mid-term diseases, hospitalization, disability, and death [11,12,13]. Furthermore, physical inactivity appears to be associated with a higher risk of dementia, Alzheimer’s disease, or mild cognitive impairment [14].

In addition to insufficient physical activity, older adults are at high risk for deficient protein intake due to factors, such as comorbidity, loss of appetite, poor oral health, the loss of autonomy, lack of economic resources, or limited access to medical and allied health services [15,16]. Deficits in protein intake decreases health-related quality of life and accelerate age-related muscle mass waste and the development of sarcopenia [17]. Moreover, malnutrition seems to be related to impaired cognition and Alzheimer’s disease pathology [18]. From an economic perspective, the price of managing patients at risk of malnutrition is very cost effective, which emphasizes the implementation of strategies focused on preventing patients from becoming malnourished [19].

One of the most promising nutritional supplements for the preservation of muscle mass in old age is beta-hydroxy-beta-methylbutyrate (HMB), a bioactive metabolite formed from the decomposition of leucine, an essential branched-chain amino acid [20,21]. HMB plays a key nutritional role as it is considered the most important regulator of muscle protein anabolism, due to its ability to stimulate the mechanistic Target of Rapamycin (mTOR) signaling pathway, which increases protein synthesis, and attenuates the proteasome pathway, inducing muscle protein catabolism [22,23]. Daily HMB supplementation (typically 3 g/day) is demonstrated to have an anti-catabolic effect, enhance protein synthesis, attenuate proteolysis, increase muscle mass, and decrease muscle damage in older adults [20,24,25]. Furthermore, animal models have recently suggested that HMB could be effective in mitigating age-related cognitive deficits [26,27] and improve the aging neuromuscular system [28]. The optimal dose of HMB cannot be obtained from a standard diet given the low quantities of HMB available in foods and the low conversion rate of leucine to HMB (~5–10%) [29]. Of further concern, HMB conversion appears to be reduced with age [30]. Thus, HMB oral supplementation stands as a realistic alternative to palliate metabolic diseases, muscle wasting, and functional loss in older adults.

The implementation of preventive strategies focused on physical and cognitive health maintenance for frail people through exercise and proper nutrition are required to contribute to lifelong wellbeing and reduce the extra costs related to physical inactivity and malnutrition. Evidence indicates that exercise programs and HMB supplementation appear to be effective and affordable strategies as independent treatments. However, there is very little evidence on the true benefits of combining both strategies. Therefore, the aim of this meta-analysis was to quantify the effects of exercise in addition to HMB supplementation on physical and cognitive health in older adults.

## 2. Materials and Methods 

### 2.1. Search Strategy

We carried out the review in accordance with a protocol that was registered in PROSPERO (Provisional ID: 147419). The systematic review was conducted according to the PRISMA (Preferred Reporting Items for Systematic Reviews and Meta-Analyses) statement [31]. A compiled PRISMA checklist is included in Table 1. A literature search was conducted by electronic search for original papers of three literature databases (Web of Science, Scopus, and PubMed). The search included original papers written in any language and published before 5 June 2019. Except for minor variations regarding database mechanisms, we used the same search string in all the databases (Table 2).

### 2.2. Inclusion Criteria

Screening and eligibility of studies were performed independently by two investigators. Discrepancies were settled by negotiation with a third author. The PICOS (population, interventions, comparators, outcomes, study design) criteria for the eligibility of studies [32] was used to determine the inclusion and exclusion criteria, as follows:Participants: Studies of participants aged 50 or older. We made no restrictions of participants’ gender, health and socio-economic status, ethnicity or geographical area. We emphasized searching for studies including people with frailty, sarcopenia, cachexia, or muscle weakness.Intervention: Any intervention combining physical exercise in addition to HMB oral supplementation. We considered every exercise activity requiring increased energy output without taking into account frequency or intensity. We considered any HMB dosage, supplementation form (powders, pills, nutritional drink) and nature (free acid or enriched).Comparators: Participants not provided with HMB supplementation (controls or placebo).Outcomes: Clinical outcomes on physical and cognitive health, including (but not restricted to) changes in physical function, muscular strength, body composition, cognitive impairment, and quality of life.Study designs: Randomized controlled trials (RCT) were included in order to determine if the HMB oral supplementation (investigational treatment) was more effective than a control or placebo group when provided during a physical exercise program. The comprehensive search of RCT was set to identify gaps in the current evidence.

### 2.3. Data Collection

The data in the studies were evaluated by one investigator using a predefined data sheet. The extraction was checked independently by two other authors. First, all potential papers were downloaded in the citation software EndNote; second, all duplicates were deleted; third, titles and abstracts were screened to identify studies that potentially met the eligibility criteria; fourth, full texts were subsequently assessed for eligibility. Additionally, we hand-searched the reference lists of eligible papers and of several recently published reviews for further studies. Disagreements were resolved through discussions with the reviewers. For our meta-analyses, we collected the following data for both the exercise groups and control groups: Group sizes, and the mean differences of selected outcomes (after–before) with a 95% CI or SD.

### 2.4. Quality Assessment and Risk of Bias

Risk of bias was assessed independently by two investigators using the latest version of the Cochrane Collaboration Risk-of-Bias tool (RoB, 2 March 2019) in randomized trials [33]. Studies were assessed in five domains: Bias arising from the randomization process; bias due to deviations from intended interventions; bias due to missing outcome data; bias in measurement of the outcome; bias in selection of the reported result. The tool includes algorithms that map responses to signaling questions onto a proposed risk-of-bias judgement for each domain in three levels: Low risk of bias, some concerns, and high risk of bias.

### 2.5. Data Analysis

The effect sizes (ESs) were calculated as the standardized mean differences between the HMB supplementation and placebo groups. The sample size and mean ES across all studies were used to calculate the variance around each ES. Meta-analyses were performed using robust variance estimation (RVE) with small-sample corrections. RVE is a form of random-effects meta-regression for multilevel data structures, which allows for multiple effect sizes from the same study to be included in a meta-analysis, even when information on the covariance of these effect sizes is unavailable. Instead, RVE estimates the variance of meta-regression coefficient estimates using the observed residuals. It does not require distributional assumptions and does not make any requirements on the weights [34,35]. Study was used as the clustering variable to account for correlated effects within studies. Observations were weighted by the inverse of the sampling variance. A sensitivity analysis, using alternative correlational values to calculate the standard error, revealed that the choice of correlational value did not impact the overall results of the meta-analysis. I2 was used to evaluate between-study heterogeneity. Values of I2 more than 25%, 50%, and 75% were selected to reflect low, moderate, and high heterogeneity, respectively. All analyses were performed using packages robumeta (version 2.0) and metafor (version 2.0-0) in R version 3.4.4 (The R Foundation for Statistical Computing, Vienna, Austria).

## 3. Results

### 3.1. Characteristics of Studies

Out of the 936 publications from the database search, we included 10 RCTs [36,37,38,39,40,41,42,43,44,45] on physical activity and the additional effect of HMB on several measures in the final analysis. Figure 1 shows the PRISMA flow diagram. Across the 10 studies, we extracted data from 384 participants, all over 50 years of age. The majority of studies (n = 8) included healthy people. The ethnicity of the subjects was not mentioned in any study. HMB dosage varied between 1.0 (n = 1), 1.5 (n = 2), and 3.0 g/d (n = 7). HMB supplementation was administered in its calcium salt form (Ca-HMB) in nine studies, whilst only one provided the free acid form (HMB-FA). Exercise interventions lasted from 3 to 24 weeks, with a frequency between 1 to 3 days a week. HMB administration was the same as the exercise intervention except for one study [40], in which participants consumed HMB 5 days prior to bed rest and was continued until the end of the rehabilitation period. All the interventions included resistance exercises, but the routine and intensity of the programs differed. A summary of the studies included is presented in Table 3.

### 3.2. Quality of Studies and Risk of Bias

No study was considered as low risk of bias in all categories. The greatest biases were found in the concealment, randomization, and selection of the reported results. Two studies showed high risk of bias in the selection of the reported results due to important discrepancies with the pre-registered trial. Four studies did not provide a trial pre-registration or publication. Two studies showed partial results from the pre-specified plan. A summary of the risk of bias assessment is shown in Table 4.

### 3.3. Studies’ Outcomes and Results

Seven studies included data from the mean differences between control and HMB groups after exercise training on different health measures, showing controversial results (Table 5). Body composition was the most studied outcome [37,38,39,40,41,43,44], followed by muscular strength [37,38,40,43,44] and physical performance [37,40,43,44]. No study included cognitive outcomes. Studies shared 11 out of the 40 measures analyzed. Body composition was examined using dual-energy X-ray absorptiometry (DXA), bioelectrical impedance analysis (BIA), and computed tomography (CT). Two studies [43,44] found positive effects in fat free mass using different techniques, whilst three studies [37,40,41] found no differences between HMB and controls. HMB supplementation had no effects on fatty mass in absolute terms when using DXA and BIA exams [41,43,44]; in turn, the % of body fat loss after HMB supplementation increased when using skinfold analysis [41]. Two studies [37,43] examined the abdominal fat mass with contradictory results. Muscular strength included handgrip, knee flexion/extension by isokinetic, isometric and one maximum repetition (1RM) measures, and bench/leg press exercises. Out of the five RCTs exploring muscular strength, only one study [37] found positive effects of HMB supplementation in comparison with controls. Physical performance was tested using the short physical performance battery (SPPB) and its three component parts (sit-to-stand, gait speed 6 m, and get-up-and-go tests). No treatment effects were observed between exercise alone or combined with HMB supplementation in any physical performance measure except for the 6-min walking test [37].

### 3.4. Meta-Analyses

Out of all the meta-analyses, handgrip strength (Figure 2) was the only outcome close to showing statistical significance but with a small effect size (ES = 0.19 (95% CI: −0.03 to 0.40) *p* = 0.067). This result almost significantly favors the HMB supplementation against placebo with the smallest heterogeneity possible (I2 = 0%). On the other hand, the effect of HMB was not significantly harmful to leg strength (ES = −0.78 (95% CI: −3.16 to 1.59) *p* = 0.291). However, there was very high heterogeneity, I2 = 91.6% (Figure 3). Almost no effect was found in muscle mass (ES = 0.07 (95% CI −0.69 to 0.82_ *p* = 0.833, I2 = 90.6) (Figure 4). A positive non-significant effect of HMB was found in fat mass (ES = 0.61 (95% CI −0.73 to 1.96) *p* = 0.293, I2 = 84.1) (Figure 5). When muscle and strength were calculated together, HMB did not have any effect (ES = −0.06 (95% CI: −0.82 to 0.71) *p* = 0.853, I2 = 85.8) (Figure 6).

## 4. Discussion

The results from this review with meta-analysis suggest that HMB supplementation in addition to physical exercise has no or fairly low impact in improving body composition, muscle strength, or physical performance in adults aged 50 to 80 years compared to exercise alone. These findings reinforce the effectiveness of supervised and controlled exercise, alone or combined with HMB supplementation, to enhance health and functionality in older adults. Whereas the nutritional supplementation strategy was very similar among studies (3 g/d of HMB), the heterogeneity of the exercise programs (type, frequency, volume, and intensity) render the comparison between interventions difficult, which encourages replication. Finally, we identified an important gap in the literature relating to the combination of HMB and exercise and its impact to reduce cognitive impairment, as well as limited studies examining physical performance variables and including frail people, over 80 years and with very limited or no mobility.

Previous reviews have evidenced that the oral supplementation of HMB is an effective nutritional therapy to mitigate the decline in muscle mass and preserve muscle function in older adults and frail people, especially during hospital rehabilitation and recovery [20,24,25,46,47]. However, to the best of our knowledge, this is the first time comparing the impact of HMB as a nutritional strategy to optimize physical exercise interventions. Our findings revealed no critical differences in favor of HMB supplementation compared to placebo in muscle mass and strength improvements, when provided during an exercise training program. Whereas this does not contradict the promising benefits of HMB supplementation to improve lean muscle mass and preserve muscle strength in older adults, it seems to indicate that physical exercise may produce similar—and likely more—benefits to improve muscle and strength in healthy older people, especially if including a properly designed resistance training program [48,49]. 

The length of studies varied from 3 to 24 weeks. The majority of interventions included resistance training as the main part of the exercise program and lasted 8 weeks with a frequency of 2 to 3 sessions per week. Interestingly, from the five studies showing positive effects of HMB supplementation (Table 3), one was the shortest in duration (40 ± 20 days), but the one with the highest training frequencies (5 days a week) [44], another was the longest (24 weeks) [43], and the other three included an 8-week program, two with 2 sessions/week [37,41] and one with 3 sessions per week after a bed rest period [39]. While all these studies provided an optimal HMB dosage and met the general recommendations of resistance training in older adults, the particular adaptations in the program (frequency, intensity, and exercise modifications) to specific individual needs and capabilities of each older adult (e.g., frailty, mobility limitations, or osteoporosis) may account for the greater effects on health [48,49].

It is important to note that there is very limited information relating to the effects of combining HMB and exercise in frail people or with reduced mobility. In the current review, only two studies explored people with physical limitations, such as hip fracture [44] and non-cystic fibrosis bronchiectasis [36], both showing greater muscle and strength improvements after HMB supplementation. The fact that one short intervention on patients with a hip fracture found benefits in body composition after less than 6 weeks of a high frequency exercise intervention may indicate a potential positive effect of acute HMB supplementation during rehabilitation programs to help old people in recovering functional capacity after a fall. Previous interventions in older adults who underwent orthopedic surgery found that HMB accelerated wound healing, reduced dependence on bed and immobilization period, and increased muscle strength [50]. Furthermore, a pilot non-randomized control trial on sarcopenic patients with gastric cancer found that a preoperative home-based, daily exercise program (handgrip training, walking, and resistance training), with nutritional support including HMB, reduced sarcopenia and postoperative complications [51]. HMB has antioxidant and anti-inflammatory properties that ameliorate muscle loss by stimulating protein synthesis and by decreasing proteolysis [20,52,53]. Thus, it seems that HMB supplementation in addition to exercise would be particularly effective to optimize physical rehabilitation treatments and accelerate mobility recovery in frail people and muscle wasting conditions. Although evidence supporting the positive impact of HMB to enhance exercise training adaptations and increase health benefits in frail or sarcopenic people is lacking, the results of ongoing clinical trials [54,55,56] will likely provide additional evidence in the near future to support treatment choices for older people.

We found no study examining the effects of HMB and exercise to reduce older adults’ cognitive impairment. Since the appearance of HMB in the brain was detected [57], there is growing interest in its beneficial cognitive effects in aging [26,27,28,58,59,60]. Animal models suggest that HMB promotes neurite outgrowth in vitro, which is related to neuronal survival and differentiation [60], and ameliorates age-related cognitive deficits [27,58,59]. These findings encourage ongoing research into the benefits of HMB and its mechanism of action on the neuromuscular system in aging [28].

This meta-analysis has some limitations that should be noted. Although overall the quality of the studies was sufficient, some trials included in this review were at some risk of bias and should be treated with caution. Only one study examined a women sample, with the majority including both men and women with no gender distinction, which makes comparisons impossible. Furthermore, the studies included employed a variety of measures and techniques, which may provide different results and, thus, make comparisons between data sets difficult, encouraging replication. Nevertheless, all of them used standard protocols and high-quality equipment. While all the studies provided a similar dosage of HMB and included resistance training as a main part of the exercise interventions, there was substantial differences in the type, frequency, volume, and intensity of the programs. Future studies should investigate the effects of HMB and exercise through tailored, evidence-based resistance training for older adults, with a particular interest in adapted programs for people with frailty, mobility limitations, cognitive impairment, or other chronic conditions.

## 5. Conclusions

Physical exercise and HMB supplementation are two effective treatments to reduce muscle wasting and maintain or improve muscle mass in older people. This is the first systematic review and meta-analysis of RCTs examining the health benefits of combining HMB in addition to physical training in older adults. HMB seems to produce no extra improvements to exercise in physical performance, muscular strength and body composition in people aged 50 to 80 years. There is no study to date exploring the beneficial effects of HMB combined with exercise to preserve cognitive functions in aging and age-related neurodegenerative diseases. Future RCTs are needed to refine these findings to particular populations, ages, and health status. Specifically, results from interventions of HMB combined with exercise in older people with frailty, limited mobility, and/or cognitive impairment are required.

## Figures and Tables

**Figure 1 nutrients-11-02082-f001:**
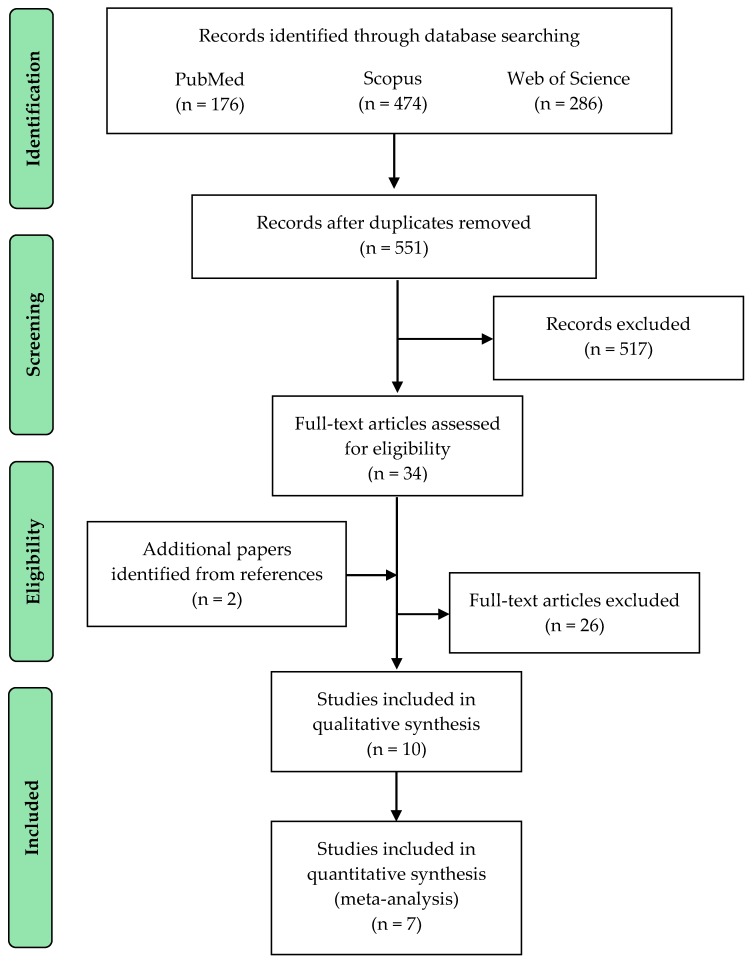
Flowchart illustrating the different phases of the search and study selection, according to the PRISMA (Preferred Reporting Items for Systematic Reviews and Meta-Analyses) statements.

**Figure 2 nutrients-11-02082-f002:**
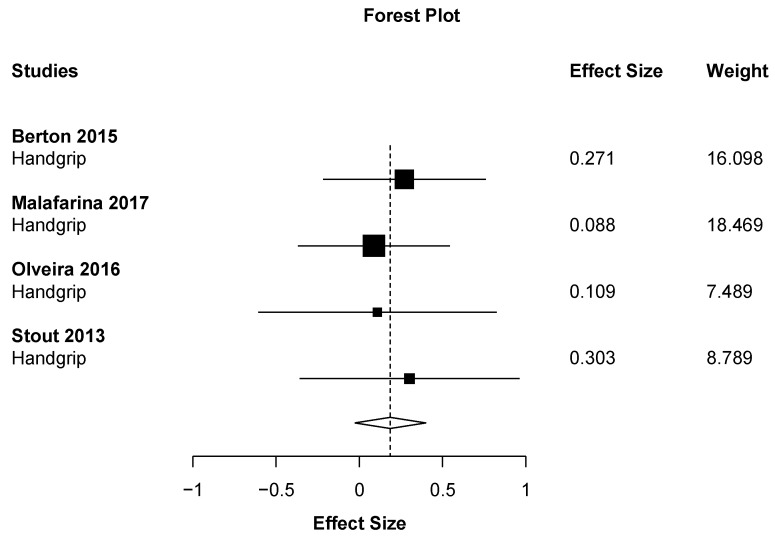
Effects meta-analysis of HMB on handgrip strength. Squares are effect sizes. The area of each square is proportional to the study’s weight in the meta-analysis. Vertical line and diamond indicate the overall measure of effects and confidence intervals.

**Figure 3 nutrients-11-02082-f003:**
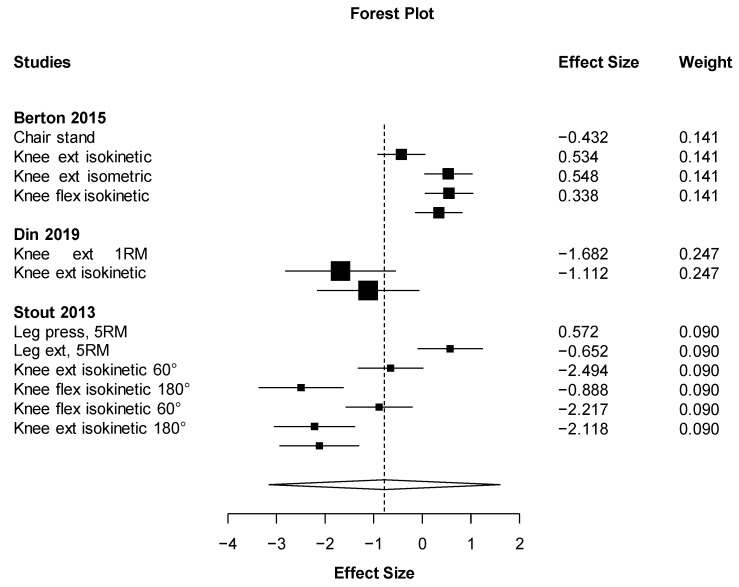
Effects meta-analysis of HMB on leg strength. Squares are effect sizes. The area of each square is proportional to the study’s weight in the meta-analysis. Vertical line and diamond indicate the overall measure of effects and confidence intervals.

**Figure 4 nutrients-11-02082-f004:**
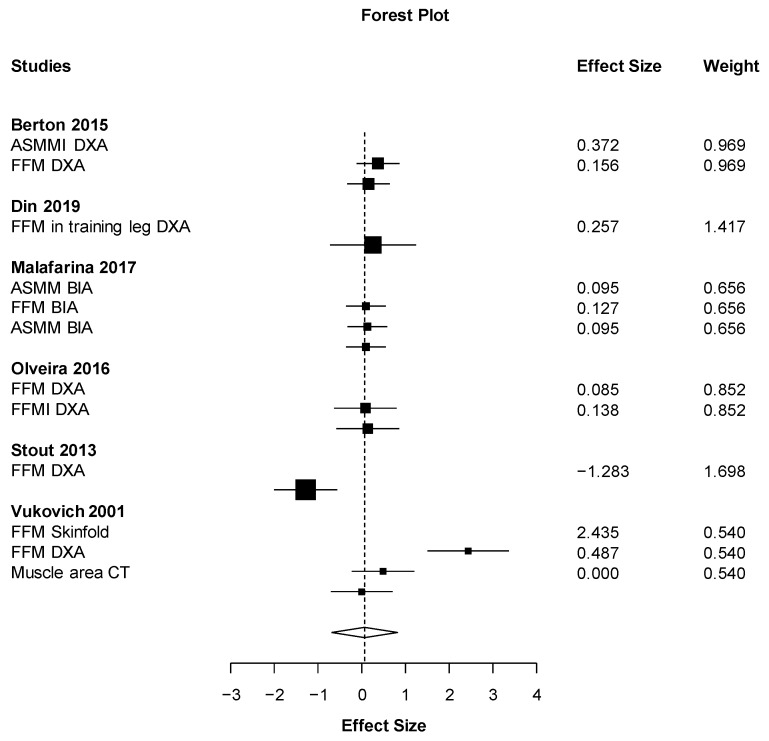
Effects meta-analysis of HMB on muscle mass. Squares are effect sizes. The area of each square is proportional to the study’s weight in the meta-analysis. Vertical line and diamond indicate the overall measure of effects and confidence intervals.

**Figure 5 nutrients-11-02082-f005:**
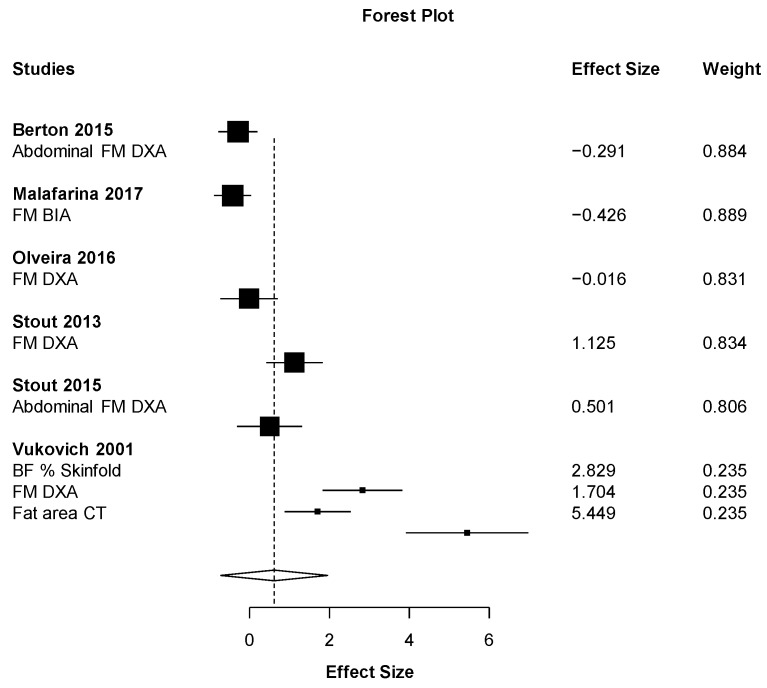
Effects meta-analysis of HMB on fat mass. Squares are effect sizes. The area of each square is proportional to the study’s weight in the meta-analysis. Vertical line and diamond indicate the overall measure of effects and confidence intervals.

**Figure 6 nutrients-11-02082-f006:**
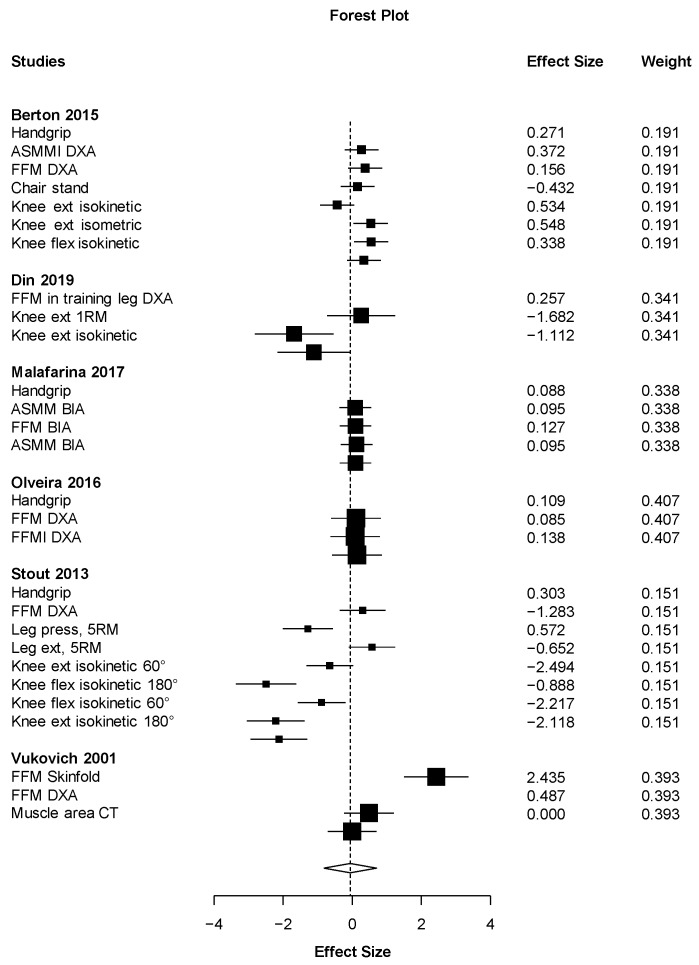
Effects meta-analysis of HMB on muscle mass and strength. Squares are effect sizes. The area of each square is proportional to the study’s weight in the meta-analysis. Vertical line and diamond indicate the overall measure of effects and confidence intervals.

**Table 1 nutrients-11-02082-t001:** Checklist of items to include when reporting a systematic review or meta-analysis.

Section/Topic	Item	Checklist Item	Page
**TITLE**			
Title	1	Identify the report as a systematic review, meta-analysis, or both.	1
**ABSTRACT**			
Structured summary	2	Provide a structured summary including, as applicable: background; objectives; data sources; study eligibility criteria, participants, and interventions; study appraisal and synthesis methods; results; limitations; conclusions and implications of key findings; systematic review registration number.	1
**INTRODUCTION**			
Rationale	3	Describe the rationale for the review in the context of what is already known.	2
Objectives	4	Provide an explicit statement of questions being addressed with reference to participants, interventions, comparisons, outcomes, and study design (PICOS).	2
**METHODS**			
Protocol and registration	5	Indicate if a review protocol exists, if and where it can be accessed (e.g., Web address), and, if available, provide registration information including registration number.	2
Eligibility criteria	6	Specify study characteristics (e.g., PICOS, length of follow-up) and report characteristics (e.g., years considered, language, publication status) used as criteria for eligibility, giving rationale.	4
Information sources	7	Describe all information sources (e.g., databases with dates of coverage, contact with study authors to identify additional studies) in the search and date last searched.	3
Search	8	Present full electronic search strategy for at least one database, including any limits used, such that it could be repeated.	3
Study selection	9	State the process for selecting studies (i.e., screening, eligibility, included in systematic review, and, if applicable, included in the meta-analysis).	3
Data collection process	10	Describe method of data extraction from reports (e.g., piloted forms, independently, in duplicate) and any processes for obtaining and confirming data from investigators.	4
Data items	11	List and define all variables for which data were sought (e.g., PICOS, funding sources) and any assumptions and simplifications made.	4
Risk of bias in individual studies	12	Describe methods used for assessing risk of bias of individual studies (including specification of whether this was done at the study or outcome level), and how this information is to be used in any data synthesis.	5
Summary measures	13	State the principal summary measures (e.g., risk ratio, difference in means).	5
Synthesis of results	14	Describe the methods of handling data and combining results of studies, if done, including measures of consistency (e.g., I^2^) for each meta-analysis.	5
Risk of bias across studies	15	Specify any assessment of risk of bias that may affect the cumulative evidence (e.g., publication bias, selective reporting within studies).	5
Additional analyses	16	Describe methods of additional analyses (e.g., sensitivity or subgroup analyses, meta-regression), if done, indicating which were pre-specified.	5
**RESULTS**			
Study selection	17	Give numbers of studies screened, assessed for eligibility, and included in the review, with reasons for exclusions at each stage, ideally with a flow diagram.	6
Study characteristics	18	For each study, present characteristics for which data were extracted (e.g., study size, PICOS, follow-up period) and provide the citations.	7
Risk of bias within studies	19	Present data on risk of bias of each study and, if available, any outcome level assessment (see item 12).	6
Results of individual studies	20	For all outcomes considered (benefits or harms), present, for each study: (a) simple summary data for each intervention group (b) effect estimates and confidence intervals, ideally with a forest plot.	5
Synthesis of results	21	Present results of each meta-analysis done, including confidence intervals and measures of consistency.	9
Risk of bias across studies	22	Present results of any assessment of risk of bias across studies (see Item 15).	7
Additional analysis	23	Give results of additional analyses, if done (e.g., sensitivity or subgroup analyses, meta-regression (see item 16)).	9
**DISCUSSION**			
Summary of evidence	24	Summarize the main findings including the strength of evidence for each main outcome; consider their relevance to key groups (e.g., healthcare providers, users, and policy makers).	13
Limitations	25	Discuss limitations at study and outcome level (e.g., risk of bias), and at review-level (e.g., incomplete retrieval of identified research, reporting bias).	14
Conclusions	26	Provide a general interpretation of the results in the context of other evidence, and implications for future research.	15
**FUNDING**			
Funding	27	Describe sources of funding for the systematic review and other support (e.g., supply of data); role of funders for the systematic review.	15

**Table 2 nutrients-11-02082-t002:** Search results from electronic databases.

Database	Keywords	Records
PubMed	Search (((((HMB)[Title/Abstract] OR beta-hydroxy-beta-methylbutyrate)[Title/Abstract] OR β-hydroxy-β-methylbutyrate[Title/Abstract]))) AND (((((elder *) OR elderly)) OR (((exercise *) OR intervention *) OR training *)) OR ((((sarcopen *) OR frail *) OR cachexia) OR “muscle weakness”))	176
Scopus	((TITLE-ABS-KEY (β-hydroxy-β-methylbutyrate) OR TITLE-ABS-KEY (hmb) OR TITLE-ABS-KEY (beta-hydroxy-beta-methylbutyrate) OR TITLE-ABS-KEY (b-hydroxy-b-methylbutyrate))) AND (((TITLE-ABS-KEY (elder *) OR TITLE-ABS-KEY (“old * adult *”))) OR ((TITLE-ABS-KEY (sarcopen *) OR TITLE-ABS-KEY (frail *) OR TITLE-ABS-KEY (cachexia) OR TITLE-ABS-KEY (“muscle weakness”))) OR ((TITLE-ABS-KEY (exercise *) OR TITLE-ABS-KEY (intervention *) OR TITLE-ABS-KEY (training *))))	474
Web of Science	TOPIC: (β-Hydroxy-β-Methylbutyrate) OR TOPIC: (hmb) OR TOPIC: (beta-hydroxy-beta-methylbutyrate) OR TOPIC: (b-hydroxy-b-methylbutyrate) AND ((TOPIC: (elder *) OR TOPIC: (“old * adult *”)) OR (TOPIC: (sarcopen *) OR TOPIC: (frail *) OR (TOPIC: (cachexia) OR TOPIC: (“muscle weakness”)) OR (TOPIC: (exercise *) OR TOPIC: (intervention *) OR TOPIC: (training *)))	286

* Broadens the search by finding words that start with the same letters.

**Table 3 nutrients-11-02082-t003:** Studies included in the analyses.

Study	Length	Age	Sample	Participants	Supplementation	Compliance	SAEs	Control	Exercise
Berton (2015)	8 weeks	69.5 (5.3)	EG = 32 CG = 34	Healthy women	1.5 g/d Ca-HMB in Ensure Plus Advance enriched with 25(OH)D 227 IU/100 mL	HMB: 96 ± 6% Exercise: N.R.	Abdominal pain, constipation (n = 2) and itching (n = 1)	Standard diet	2 × a week, mild fitness program at public gyms. Aerobic exercises to improve speed of muscle contraction, and a small part dedicated to resistance exercises, essentially to improve handgrip strength
Din (2019)	6 weeks	68.5 (1.1) ^a^	EG = 8 CG = 8	Healthy men	1.0 g/d HMB-FA in BetaTOR^®^	HMB: 99% Exercise: N.R.	N.R.	Placebo	3 × a week, supervised unilateral progressive resistance training. Leg extension of the dominant leg (6 sets, 8 rep, 75% 1-RM, adjusted each 10 days)
Malafarina (2017)	42.3 ± 20.9 days	85.4 (6.3)	EG = 49 CG = 43	Patients with a hip fracture 73.8% women	3.0 g/d Ca-HMB in Ensure Plus Advance enriched with 25(OH)D 227 IU/100 mL	HMB: >80% Exercise: N.R.	N.R.	Standard diet	5 × a week, 50-min supervised rehabilitation therapy. Exercises to strengthen the lower limbs, balance exercises, and walking re-training in individual or group
Olveira (2015)	12 weeks	56.1 (1.3)	EG = 15 CG = 15	Patients with non-cystic fibrosis bronchiectasis 60% women	1.5 g/d Ca-HMB in Ensure Plus Advance enriched with 25(OH)D 227 IU/100 mL	HMB: N.R. Exercise: 100%	N.R.	Standard diet	2 × a week, 60-min supervised exercise program at a hospital and 1 × 30-min unsupervised session. Cycle ergometer and treadmill (30 min, 75–80% VO_2_ max), upper and lower limb strength (8 min, 1 set, 8–10 rep), breathing retraining (15 min), and stretching and relaxation (7 min)
Stout (2013) †	24 weeks	73.0 (1.0) ^a^	EG = 16 CG = 20	Healthy older adults 54.2% women	3.0 g/d Ca-HMB + 8 g/d carbohydrate	HMB: >67% Exercise: >60%	N.R.	Placebo	3 × a week, supervised resistance training. Bench press, leg press, leg extension (1–3 sets, 8–12 rep, 80% 1RM, adjusted), lat pulldown hack squat (1–3 sets, 8–12 rep, 2–5 min rest)
Stout (2015) †	12 weeks	72.1 (5.7)	EG = 12 CG = 12	Healthy men	3.0 g/d Ca-HMB + 8g/d carbohydrate	HMB: >67% Exercise: >60%	N.R.	Placebo	3 × a week, supervised resistance training. Bench press, leg press, leg extension (1–3 sets, 80% 1RM, adjusted), lat pulldown hack squat (1–3 sets, 8–12 rep, 2–5 min rest)
Vukovich (2001)	8 weeks	70 (1.0)	EG = 14 CG = 17	Healthy older adults 54.6% women	3.0 g/d Ca-HMB	HMB: 100% Exercise: 100%	No adverse reaction or medical complication	Placebo	2 × a week, supervised resistance training and 3 × walking (40 min self-paced) and stretching (10 min). Overhead press, bench press, l at pulldown, elbow extension and flexion, leg flexion/extension, and leg press (2 sets, 8–12 reps. 70% 1RM, adjusted each 2 weeks)
**After bed rest**									
Deutz (2013) * Standley (2017) *	8 weeks	67.4 (1.4) ^a^	EG = 11 CG = 8	Healthy older adults 78.9% women	3.0 g/d Ca-HMB	HMB >67% Exercise >60%	No serious adverse events	Placebo	3 × a week, 1-h resistance training rehabilitation after a 10-day bed rest. 1-h circuit training for combined hip and knee extensors and flexors, light upper body exercises (3 sets, 8–10 rep, 80% 1RM) and self-paced walking
**Results were not showed separately for old people**									
Nissen (2000)	8 weeks	63–81 ^b^	EG = 18 CG = 18	Healthy older adults	3.0 g/d Ca-HMB	N.R.	Less diarrhea and less loss of appetite	Placebo	3 × a week, supervised resistance training. Alternated exercising of either the upper or lower body during each exercise session
	8 weeks	62–79 ^b^	EG = 16 CG = 18	Healthy older adults	3.0 g/d Ca-HMB	N.R.	Less diarrhea and less loss of appetite	Placebo	2 × a week resistance training + 3 × 60-min walking and stretching

^a^ Mean age of the whole sample was not reported; therefore, the mean age of the experimental group is presented; ^b^ Range of the experimental group; * the same population; † part of the same population; EG: experimental group; CG: control group; Ca-HMB: calcium beta-hydroxy-beta-methylbutyrate; HMB-FA: beta-hydroxy-beta-methylbutyrate free acid SAEs: serious adverse events; N.R.: not reported.

**Table 4 nutrients-11-02082-t004:** Risk of bias of included studies.

	Randomization Process	Deviations from Intended Interventions	Missing Outcome Data	Measurement of the Outcome	Selection of the Reported Result	Overall Bias
Berton (2016)	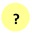					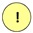
Deutz (2013)			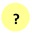		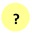	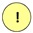
Din (2019)					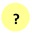	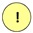
Malafarina (2017)	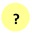	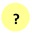	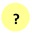		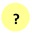	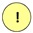
Olveira (2015)	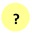	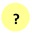			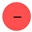	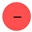
Standley (2017)					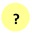	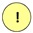
Stout (2013)	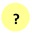				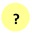	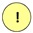
Stout (2015)	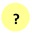				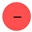	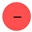
Vukovich (2001)	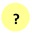				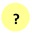	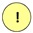
Nissen (2000)					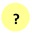	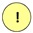

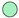
 Low risk of bias; 
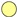
 Unclear risk of bias; 
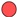
 High risk of bias.

**Table 5 nutrients-11-02082-t005:** Effect of beta-hydroxy-beta-methylbutyrate (HMB) on health parameters.

Outcome	Measure	Overall effect *	Study
Physical performance	SPPB	No effect	Berton (2015)
		No effect ^a^	Deutz (2013)
	6-min walking test	**Positive**	Berton (2015)
	Gait speed	No effect	Malafarina (2017)
	Get-up-and-go	No effect	Stout (2013)
		No effect ^a^	Deutz (2013)
Muscular strength	Isokinetic knee flexion	**Positive**	Berton (2015)
		No effect	Stout (2013)
	Isokinetic knee extension	**Positive**	Berton (2015)
		No effect	Stout (2013)
		No effect	Din (2019)
		No effect ^a^	Deutz (2013)
	Isometric knee extension	**Positive**	Berton (2015)
	Handgrip strength	No effect	Berton (2015)
		No effect	Malafarina (2017)
		No effect	Stout (2013)
	Handgrip strength endurance	**Positive**	Berton (2015)
	Handgrip work index	No effect	Malafarina (2017)
	Knee extension, 1RM	No effect	Din (2019)
	Bench press, 5RM	No effect	Stout (2013)
	Leg press, 5RM	No effect	Stout (2013)
	Leg extensor, 5RM	No effect	Stout (2013)
Body composition	Fat free mass (DXA)	No effect	Berton (2015)
		**Positive** ^b^	Stout (2013)
		No effect	Vukovich (2001)
		No effect ^a^	Deutz (2013)
	Fat free mass (BIA)	**Positive**	Malafarina (2017)
	Fat free mass (Skin fold thickness)	No effect	Vukovich (2001)
	ASMMI	No effect	Berton (2015)
	Muscle mass (BIA)	**Positive**	Malafarina (2017)
	Appendicular lean mass (BIA)	**Positive**	Malafarina (2017)
	Skeletal muscle mass (BIA)	No effect	Malafarina (2017)
	ASMM (BIA)	**Positive**	Malafarina (2017)
	Fatty mass (DXA)	No effect	Stout (2013)
	Fatty mass (BIA)	No effect	Malafarina (2017)
	Fatty mass % (Skin fold thickness)	**Positive**	Vukovich (2001)
	Fatty mass % (DXA)	No effect	Vukovich (2001)
	Leg lean mass (DXA)	No effect	Stout (2013)
		No effect ^a^	Deutz (2013)
	Arm lean mass (DXA)	**Positive** ^b^	Stout (2013)
	Abdominal fat mass (DXA)	No effect	Berton (2015)
		**Positive**	Stout (2015)
	Radial muscle density (CT)	**Positive**	Berton (2015)
	Radial muscle area (CT)	No effect	Berton (2015)
	Radial fat area (CT)	No effect	Berton (2015)
	Radial fat/muscle ratio (CT)	**Positive**	Berton (2015)
	Tibial muscle density (CT)	**Positive**	Berton (2015)
	Tibial muscle area (CT)	No effect	Berton (2015)
	Tibial fat area (CT)	No effect	Berton (2015)
	Tibial fat/muscle ratio (CT)	No effect	Berton (2015)
	Fat area (CT)	**Positive**	Vukovich (2001)
	Muscle area (CT)	No effect	Vukovich (2001)
	Cross-sectional area (VLB)	No effect ^a^	Standley (2017)
	Thigh lean mass (DXA)	No effect	Din (2019)
	Vastus lateralis thickness (DXA)	No effect	Din (2019)
Others	Muscle quality (Isokinetic knee extension 60°)	No effect	Stout (2013)
	Muscle quality (Isokinetic knee extension 180°)	No effect	Stout (2013)
	Muscle quality (Handgrip strength)	No effect	Stout (2013)
	Proteins expression (histology)	**Positive** ^a^	Standley (2017)

* HMB effect compared to CG (*p* < 0.05); ^a^ Bed rest + rehabilitation; ^b^ male only; RM—Repetition maximum; MVC—Maximal voluntary contraction; ASMMI—Appendicular skeletal muscle mass index; ALM—Appendicular lean mass; ASMM—Appendicular skeletal muscle mass; Muscle quality—Muscle strength relative to muscle mass; DXA—Dual X-ray absorptiometry; CT—Computed tomography; BIA—Bioelectrical impedance analysis; VLB—Vastus lateralis biopsy.

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
