# Peer review of "Health Benefits of β-Hydroxy-β-Methylbutyrate (HMB) Supplementation in Addition to Physical Exercise in Older Adults: A Systematic Review with Meta-Analysis"

_nutrients, 2019, doi:10.3390/nu11092082_

Round 1

Reviewer 1 Report

The manuscript presents a systematic review and meta-analysis of randomised controlled trial testing the impact of protein supplementation in combination with physical exercise on the physical and cognitive health of adults aged 50+. 10 studies were identified as eligible, with no observed superiority of HMB supplementation+exercise when compared with exercise only for physical status/performance. No included studies investigated effects on cognition.

This is an interesting review with a good rationale and sound analyses. As a minor limitation, there doesn't seem to be a review protocol available for this study, thus, it is not possible to ascertain that the review adheres to the authors' initial plan. My main comments are listed below:

The authors need to improve the Methods section by providing a more structured and detailed account of their inclusion criteria: To this end, I would strongly recommend using the PICOS criteria. For instance, the authors do not mention outcomes of interest in the Methods section, and do not justify the reason why only RCTs were included in the search. Using the PICOS structure would help to clarify these points. As good practice, systematic reviews should include a compiled PRISMA checklist to ensure that correct reporting standards are adhered to. In the search strategy, the authors should provide information on any time limits applied (or not) to the search. In Section 2.3 Data Collection, it is unclear how many reviewers were involved in screening abstracts and/or full-texts. Also, it is not clear whether any third party intervened to moderate disagreements. Section 3.1 - It is unclear whether the duration of the HMB administration was the same as the exercise intervention in all studies. Please clarify. Section 3.2 - It would be useful to have a summary of the risk of bias assessment in a tabular format. Line 137 mentions a supplementary figure (figure S1), but no supplementary materials appear to be present in the submission system.

Author Response

We are grateful for the positive evaluation of our manuscript, and we also very much appreciate the reviewers’ suggestions, which have been very helpful in improving the final version.

We have carefully addressed all the reviewers’ suggestions and provided a detailed point-by-point response to each comment. Changes are highlighted in colors (blue for Reviewer 1, green for Reviewer 2). The location of the corrections refers to the new version of the manuscript.

Response to Reviewer 1 Comments

Point 1. This is an interesting review with a good rationale and sound analyses. As a minor limitation, there doesn't seem to be a review protocol available for this study, thus, it is not possible to ascertain that the review adheres to the authors' initial plan. My main comments are listed below:

Response 1. We appreciate the positive comments. The reviewer is absolutely right and we are very sorry we have not received the PROPOSERO assessment on time. The preliminary ID number is 147419, with current status “being assessed by the editorial team”, so we are expecting to get the registration number in the following days.

Point 2. The authors need to improve the Methods section by providing a more structured and detailed account of their inclusion criteria: To this end, I would strongly recommend using the PICOS criteria. For instance, the authors do not mention outcomes of interest in the Methods section, and do not justify the reason why only RCTs were included in the search. Using the PICOS structure would help to clarify these points.

Response 2. Thank you for this important suggestion. We have detailed the inclusion criteria according to the PICOS structure. Please find the section 2.2. Inclusion criteria improved accordioning [L95-116]

Point 3. As good practice, systematic reviews should include a compiled PRISMA checklist to ensure that correct reporting standards are adhered to.

Response 3. We totally agree, we apologize for this missing information. Please find a compiled PRISMA checklist included in Table 1, as well as the proper citation in L74-76. We have also updated the Figure 1 (PRISMA flowchart) to distinguish between studies included in qualitative and quantitative synthesis.

Point 4. In the search strategy, the authors should provide information on any time limits applied (or not) to the search.

Response 4. The search included original papers published before 05 Jun 2019. Please find this information included in section 2.1. Search Strategy [L77]. We have also clarified that the search strategy included original papers written in any language, although eventually all the included studies were written in English.

Point 5. In Section 2.3 Data Collection, it is unclear how many reviewers were involved in screening abstracts and/or full-texts. Also, it is not clear whether any third party intervened to moderate disagreements.

Response 5. We are sorry again for this missing information. Screening and eligibility of studies were performed independently by two investigators. Discrepancies were settled by negotiation between a third author. We have included this information in the new version of the manuscript [L95-96]. We have also clarified that the data in the studies were evaluated by one investigator using a predefined data sheet and checked independently by two other authors [L121-122].

Point 6. Section 3.1 - It is unclear whether the duration of the HMB administration was the same as the exercise intervention in all studies. Please clarify.

Response 6. Thanks for this comment. Certainly, HMB administration was the same as the exercise intervention except for one experiment, in which participants consumed HMB 5 days prior to bed rest and was continued until the end of the rehabilitation period. We have clarified this information in Section 3.1. Characteristics of studies [L164-164]

Point 7. Section 3.2 - It would be useful to have a summary of the risk of bias assessment in a tabular format. Line 137 mentions a supplementary figure (figure S1), but no supplementary materials appear to be present in the submission system.

Response 7. Actually, that supplementary figure was supposed to show the risk of bias. We apologize for this mistake during the submission process. We agree with the reviewer this is a very useful information, so we have included it in the text as Table 4. 

Reviewer 2 Report

The authors have extensively analyzed the effect of Beta-hydroxy-beta-methylbutyrate (HMB) on elderly individuals. They concluded that supplementation of HMB in addition to physical exercise do not have any additional effect on the overall health of the individuals. They have identified correctly that extensive studies are needed to understand role of HMB in cognitive health of elderly people.

A few points the authors might consider:

The length of studies mentioned - varies from 3 to 24 weeks. Comment on that. Might consider adding ethnicity of the subjects in table 2. Might consider commenting on the gender bias too.

Author Response

We are grateful for the positive evaluation of our manuscript, and we also very much appreciate the reviewers’ suggestions, which have been very helpful in improving the final version.

We have carefully addressed all the reviewers’ suggestions and provided a detailed point-by-point response to each comment. Changes are highlighted in colors (blue for Reviewer 1, green for Reviewer 2). The location of the corrections refers to the new version of the manuscript.

Response to Reviewer 2 Comments

Point 1. The authors have extensively analyzed the effect of Beta-hydroxy-beta-methylbutyrate (HMB) on elderly individuals. They concluded that supplementation of HMB in addition to physical exercise do not have any additional effect on the overall health of the individuals. They have identified correctly that extensive studies are needed to understand role of HMB in cognitive health of elderly people.

Response 1. We thank the reviewer for the interest shown in our study and highlight the strengths of this research.

Point 2. The length of studies mentioned - varies from 3 to 24 weeks. Comment on that.

Response 2. Sure, this is a critical point to discuss. We have highlighted this fact as a finding of the review in the results section [L163]. Furthermore, we have included further comments on these findings throughout the discussion section [L276-283 and L288-291].

Point 3. Might consider adding ethnicity of the subjects in table 2.

Response 3. Thanks for this suggestion, definitely this is essential information. We have checked the studies and we found no one that mentioned the ethnicity of the subjects. Please find this information included in the new version of the manuscript [L160].

Point 4. Might consider commenting on the gender bias too.

Response 4. Yes, we agree. Despite the majority of studies included both women and men, they made no gender distinction, which makes comparisons impossible. In addition, only one study examined a women sample. Comments on this fact are now included in L312-313.